# GeneMamba: Early Parkinson's Detection via Wearable Device and Genetic Data

## Abstract

Parkinson's disease (PD) is a progressive neurodegenerative disorder affecting millions worldwide, with its prevalence expected to rise as the global population ages. Early diagnosis is crucial for effective management and improved quality of life for patients. However, current accelerometer-based studies focus more on detecting the symptoms of PD, while less research has been conducted on early detection of PD. This study presents a novel multi-modal deep learning model named GeneMamba for early PD diagnosis, using state space modelling approaches to effectively analyze sequences and combining accelerometer data from wearable devices with genetic variants data. Our model predicts early PD occurrence up to 7 years before clinical onset, outperforming existing methods. Furthermore, through knowledge transfer, we enable accurate PD prediction using only wearable device data, enhancing our model's real-world applicability. Additionally, our interpretation methods uncover both established and previously unidentified genes associated with PD, advancing our understanding of the disease's genetic architecture and potentially highlighting new therapeutic targets. Our approach not only advances early PD diagnosis but also offers insights into the disease's etiology, paving the way for improved risk assessment and personalized interventions.

## 1 Introduction

Parkinson's disease (PD) is a progressive neurodegenerative disorder that affects millions of individuals globally. With over 8.5 million people worldwide living with PD and approximately 90,000 people diagnosed with Parkinson's disease each year in the United States, it represents a significant public health challenge and a substantial burden on healthcare systems (Bhidayasiri et al., 2024; Willis et al., 2022). PD is characterized by motor symptoms such as tremors, rigidity, and bradykinesia, as well as non-motor symptoms including cognitive impairment, sleep disorders, and depression (Sveinbjornsdottir, 2016). As the global population ages, the prevalence of PD is expected to rise, underscoring the urgent need for improved diagnostic and treatment strategies.

Early diagnosis of Parkinson's disease allows for timely intervention, which can slow disease progression and help manage symptoms more effectively (Emamzadeh & Surguchov, 2018). By identifying at-risk individuals during the prodromal phase, healthcare providers can initiate targeted monitoring and personalized treatment plans, potentially improving long-term patient outcomes (de Bie et al., 2020). Moreover, early diagnosis enables patients and their families to better prepare for the challenges associated with the disease, including planning for future care needs and accessing support services.

Despite the importance of early detection, current diagnostic methods for PD often rely on clinical observations of motor symptoms, which typically manifest when significant neuronal loss has already occurred. This highlights the need for innovative approaches to identify PD in its preclinical or early stages.

Recent advancements have opened new avenues for early PD prediction, particularly in wearable devices and deep learning. The integration of accelerometer data and deep learning holds promise for enhancing the accuracy and timeliness of PD diagnosis. Accelerometer data, which can capture subtle changes in movement patterns, has shown potential for detecting early motor manifestations of PD (Borzì et al., 2023; Sun et al., 2021). In addition, we propose combining accelerometer

data with genetic data to identify patients at high risk of developing PD from an early stage. The incorporation of genetic data in PD prediction models offers the opportunity to uncover new insights into the disease's etiology. While several genetic variants have been found to be associated with PD risks, many aspects of the genetic architecture of the disease remain unknown. Interpretation methods applied to genetic data could help identify novel genetic variants related to PD, contributing to our understanding of the disease mechanisms and potentially revealing new therapeutic targets.

In this paper, we propose a multi-modal deep learning model, GeneMamba, that first applies Mamba to accelerometer data with cross-modality fusion for the early prediction of PD seven years before clinical onset. By integrating the diverse data sources, we not only build an early PD prediction model, predicting up to 7 years before clinical onset, but also identify novel genes related to PD, helping to identify individuals at risk in advance. Our main contributions are as follows:

1. We propose the first application of Mamba, a Structured State Space Sequence model, to accelerometer data, combined with genetic data through cross-modality fusion for early PD prediction, outperforming existing methods up to seven years before clinical onset.

2. Our model leverages cross-modality to harness diverse data sources for accurate early PD prediction. Furthermore, recognizing that genetic data are often challenging to obtain, we employ knowledge distillation to transfer insights from the complex genetic information to more accessible accelerometer data, enabling our model to maintain high prediction accuracy while using wearable device data alone, and enhancing its practical utility in developing real-world health monitoring systems for early detection and intervention.

3. Our interpretation methods reveal both existing genes related to PD and novel genes not previously identified, helping to identify individuals at higher risk of developing PD.

## 2 RELATED WORKS

PD prediction has gathered significant attention in recent years. Researchers have employed various methods and modalities for the classification and prediction of PD, ranging from neuroimaging techniques to handwriting analysis and vocal feature extraction. This section provides an overview of the current approaches in PD-related research.

Magnetic Resonance Imaging (MRI) have shown promise in PD prediction. Shu et al. (2021) extracted white matter features from structural MRI scans and combined Support Vector Machine (SVM) and logistic regression algorithms to classify between stable PD and progressive PD. Their model achieved an Area Under the Curve (AUC) of 0.836, demonstrating the potential of MRI-based features in predicting PD progression. Handwriting analysis has emerged as another valuable tool for PD prediction. Li et al. (2022) proposed a Continuous Convolution Network (CC-Net) to distinguish between healthy individuals and PD patients based on handwriting samples, with an average AUC of 0.934 and an accuracy of 0.893. Speech impairment is a common symptom in PD, making vocal feature analysis a relevant area of study. Quan et al. (2022) developed a method that extracts time series features from speech signals and processes them using time-distributed two-dimensional convolutional neural networks (2D-CNNs) and a one-dimensional CNN (1D-CNN) for PD detection. Their approach achieved an accuracy of up to 0.92 on one of the speech tasks, demonstrating the potential of vocal biomarkers in PD detection.

Wearable devices offer a non-invasive and accessible method of collecting movement data for PD prediction and symptom detection. Several studies have focused on specific PD symptoms using these devices. Freezing of Gait (FOG), a debilitating symptom commonly associated with PD, has been a focus of such studies. Borzì et al. (2023) utilized a single inertial sensor attached to the waist to collect accelerometer data and applied a multi-head convolutional neural network to predict FOG. Their model achieved an AUC of 0.946. Hand tremor is another common symptom of PD. Sun et al. (2021) proposed a method using data collected from a wrist sensor and an 8-layer convolutional neural network (CNN) to classify PD rest, postural, and action tremors. Their approach achieved an accuracy over 0.95.

While much research has focused on detecting PD and its symptoms, early prediction of PD remains a challenging and less-studied area. Schalkamp et al. (2023) addressed this gap by using accelerometer data from the UK Biobank dataset to predict PD up to seven years before clinical

diagnosis. Employing machine learning methods, they achieved a mean Area Under the Precision-Recall Curve (AUPRC) of 0.78 in distinguishing prodromal PD from matched controls. This study highlights the potential of longitudinal accelerometer data in early PD prediction.

In summary, the field of PD symptom detection and prediction has seen significant advancements across various modalities, including MRI, handwriting images, speech signals, and wearable device data. While each approach shows potential, the integration of multiple modalities and the focus on early prediction still remain areas that need further exploration. Our study bridges the gap between deep learning and early PD prediction by proposing a multi-modal model, namely GeneMamba, that can accurately predict PD seven years before its first diagnosis. Our model effectively integrates accelerometer and genetic data through Mamba and cross-modality fusion, and further incorporates a knowledge distillation approach, enabling fine-grained early PD prediction with enhanced real-world applicability. Furthermore, we employ interpretability methods to provide insights into the model's decision-making process, thus supporting more informed clinical decision-making. This methodology not only advances the field of early PD prediction research, reaching an AUPRC of 0.859 in predicting early PD, but also reveals genes that offer potential targets for therapeutic intervention and biomarker development, enabling early detection and personalized treatment strategies for individuals at risk of developing PD.

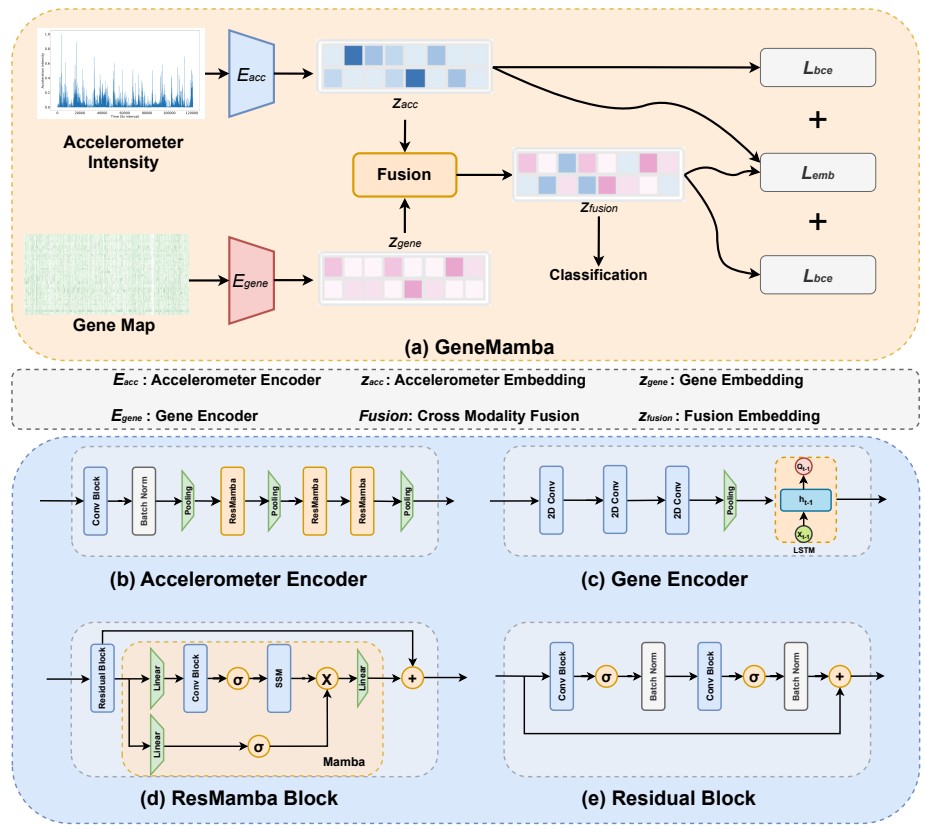

Figure 1: Architecture of GeneMamba.

## 3 METHODS

### 3.1 PRELIMINARY

**Structured State Space Model** (S4) is a framework used to represent the dynamics of a system through state variables, inputs, and outputs (Gu et al., 2022). S4 represents a system's underlying state as a vector that evolves over time according to a set of equations, while observations are

treated as functions of this state. S4 consists of two main components: a state transition equation that describes how the system's state changes from one time point to the next, and an observation equation that relates the hidden state to observable measurements. By representing the system's state at each time step and modeling how it evolves over time through state transition equations, S4 can effectively handle long-range dependencies and continuous processes inherent in time-series data, making it suitable for processing time-series data. The formula of S4 is shown below:

$$\dot{x}(t) = Ax(t) + Bu(t)$$
$$y(t) = Cx(t) + Du(t)$$

where $u(t)$ is the input vector, $x(t)$ is the state vector, $\dot{x}(t)$ represents the time derivative of the state vector, $y(t)$ is the output vector, and $A$, $B$, $C$, and $D$ are matrices that define the system dynamics.

**Mamba** is a State Space Model with a novel selection mechanism. Unlike Transformer, which rely heavily on attention mechanisms, Mamba employs a selective state space mechanism that dynamically adjusts its parameters based on the input sequence (Gu & Dao, 2023). Mamba replaces the complex attention blocks of Transformer with a state space block, leading to faster inference and lower computational complexity. Additionally, the selection mechanism makes the model not only more efficient but also capable of filtering out less relevant data, focusing on crucial information from the sequences. These designs make Mamba a promising backbone model in long sequence modeling tasks.

## 3.2 MODEL

In this study, we present a deep learning model mainly composed of three modules: the Accelerometer Encoder, the Gene Encoder, and the Fusion Module, as shown in Figure 1. The Accelerometer Encoder processes the 3-day time-series accelerometer data collected from subjects in the UK Biobank (UKBB). We propose using the Mamba model to handle the long sequence of acceleration intensity data. The Gene Encoder processes the gene variants of each subject, beginning with dimensionality reduction and feature extraction via a 1D convolutional neural network (1D-CNN) and subsequently integrating the information from individual variants using a long short-term memory (LSTM) network (Hochreiter, 1997). Finally, to merge the features from both the accelerometer data and gene variants, we introduce a Cross-Attention Fusion module (Lin et al., 2022). The outputs of this module are then passed through linear layers to compute the probability of the subject belonging to PD. Assuming $X_{acc} \in \mathbb{R}^t$, $X_{gene} \in \mathbb{R}^{n \times m}$, this process can be represented as:

$$\begin{aligned}
z_{acc} &= ACCEncoder(X_{acc}) \in \mathbb{R}^d \\
z_{gene} &= GeneEncoder(X_{gene}) \in \mathbb{R}^d \\
z_{fusion} &= Fusion(z_{acc}, z_{gene}) \in \mathbb{R}^d \\
y_{acc} &= Linear(z_{acc}) \in [0, 1] \\
y_{fusion} &= Linear(z_{fusion}) \in [0, 1]
\end{aligned} \tag{1}$$

where $z_{acc}$ and $z_{gene}$ represent the embedding vectors of the Accelerometer Encoder and Gene Encoder, respectively. $z_{fusion}$ is the output fusion vector of $z_{acc}$ and $z_{gene}$. $y_{acc}$ and $y_{fusion}$ are the prediction results from the Accelerometer Encoder and the fusion model, respectively.

**Accelerometer Encoder**

The Accelerometer Encoder processes time-series acceleration intensity data collected over a 3-day period at 5-second intervals, represented by $X_{acc} \in \mathbb{R}^t$. The initial 1D CNN layer serves to extract low-level features and significantly reduce the input dimension with a stride of 7. This dimensionality reduction is crucial for efficient processing of the time-series data, as considerable noise present in the collected data.

The initial 1D CNN layer is followed by a stack of ResMamba blocks. The ResMamba block integrates a Mamba block with a Residual block, leveraging the strengths of both architectures. The Mamba block excels at processing temporal relationships in time-series data, effectively handling long-range dependencies and focusing on crucial information. Complementing this, the Residual block is proficient in processing spatial information, aiding in noise reduction and the identification of important time periods related to the PD manifestations. By alternating these structures, the network can simultaneously learn spatial and temporal features at multiple scales, which is particularly

effective for processing the complex patterns inherent in acceleration data. We gradually increase the size of the feature embeddings from 64 to 512, allowing the network to capture increasingly abstract representations of the input data. The proposed architecture enables the network to capture and process the diverse dynamics of acceleration intensity data more comprehensively, addressing both the spatial and temporal aspects of the input while filtering out irrelevant information.

**Gene Encoder**

We propose Gene Encoder to process the genetic data obtained from the genome-wide association study (GWAS) results. The input data is a 2D matrix, represented by $X_{gene} \in \mathbb{R}^{n \times m}$, where $n$ represents the number of Single Nucleotide Polymorphisms (SNPs) identified as significant in GWAS, and $m$ represents a region of SNPs within the linkage disequilibrium (LD) range of each significant SNP.

LD is a common phenomenon in genetics where alleles at different loci are inherited together more frequently, which means that the presence of a specific allele at one SNP can predict the presence of a specific allele from another SNP if the two SNPs are in LD (Slatkin, 2021). However, the functionally relevant or associated SNPs may not necessarily be at the center of the LD region, but could occur at any position within the LD range.

To address this positional variability, we leveraged the spatial invariance property of 2D CNNs, enabling the model to detect significant patterns regardless of their location within each $m$-SNP window. We implemented this approach by treating the $n \times m$ gene map as a 2D feature map of depth 1 and setting the kernel size to $1 \times M$. The kernel size of 1 along the $n$ dimension ensures independent processing of each LD window, while $M$ accommodates the positional variability within the LD region, thereby preserving the unique information of each genetic locus. This architecture allows the CNN to effectively capture LD patterns across various positions in the genetic sequence while maintaining the ability to differentiate between individual SNPs and their associated LD regions, thus improving the model's ability to identify functionally relevant genetic variations. The 2D CNN also serves to reduce dimensionality along the $m$ axis, which is essential because LD regions often contain a considerable amount of noise and redundant information, as many genetic variants present do not have a strong association with the trait or disease of interest.

Following the CNN layers, an LSTM layer aggregates the processed genetic information into vectors. Our architecture enables Gene Encoder to effectively process patterns in genetic data, mitigating the noise while extracting and enhancing the most informative features in each LD region.

**Fusion**

The Fusion module combines embeddings from genetic and accelerometer data through two parallel Cross Attention modules. In one, gene embeddings are the query while accelerometer embeddings are the key and value, while the accelerometer embeddings serve as the query and gene embeddings serve as the key and value in the other. The result is the concatenation of the outputs of the two Cross Attention modules. This Cross Attention approach enables inter-modal information exchange, capturing complex relationships between genetic variations and physical activity patterns. By allowing each modality to selectively focus on relevant information from the other, the module outputs a comprehensive data representation, uncovering subtle interactions between genetic information and physical behaviors. The fusion process can be represented as:

$$z_{ag} = CrossAtt(q = X_{acc}, k = X_{gene}, v = X_{gene}) \in \mathbb{R}^d$$
$$z_{ga} = CrossAtt(q = X_{gene}, k = X_{acc}, v = X_{acc}) \in \mathbb{R}^d \quad (2)$$
$$z_{fusion} = Linear(Concat(z_{ag}, z_{ga})) \in \mathbb{R}^d$$

### 3.3 Loss

Our model used both genetic and accelerometer data as inputs. While accelerometer data is easily obtainable through wearable devices like smartwatches, genetic data is difficult to acquire for ordinary people. We propose to transfer the knowledge learned from Gene Encoder to the Accelerometer Encoder. This strategy offers two key advantages: first, it allows our model to achieve improved results based solely on accelerometer data, and second, it removes the need for genetic data collection, which is often unfeasible for home users. We added the knowledge transfer loss to the loss function,

and it is calculated as follows:

$$L_{\text{emb}} = \|z_{fusion} - z_{acc}\|_2^2$$
$$L_{\text{bce}} = BCE(y, y_{fusion}) + BCE(y, y_{acc})$$
$$L = \alpha L_{\text{emb}} + (1 - \alpha)L_{\text{bce}}$$
(3)

where $L_{\text{emb}}$ is the mean squared error between the accelerometer embedding and the fusion embedding. $L_{\text{bce}}$ consists of the binary cross-entropy loss of the Fusion model outputs and accelerometer model outputs compared to the ground truth, and $\alpha$ is the weight of the $L_{\text{emb}}$ loss. By aligning the embedding of the Accelerometer Encoder with the Fusion module, we gradually transfer knowledge from the Gene Encoder to the Accelerometer Encoder, not only improving the results when using accelerometer data only but also making the model suitable for scenarios where genetic data is unavailable.

## 4 DATA

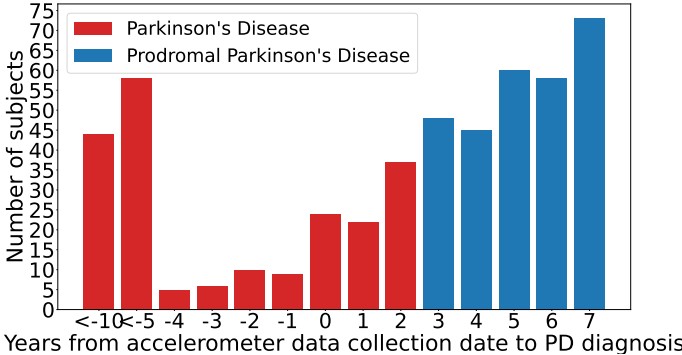

Figure 2: Distribution of the years from accelerometer data collection date to PD diagnosis.

UK Biobank (UKBB) is a large-scale biomedical database established in the United Kingdom. Established between 2006 and 2010, the project recruited around 500,000 participants, aged 40 to 69 years, from across the UK. Participants underwent initial assessments at 22 centers nationwide, providing blood, urine, and saliva samples, as well as detailed information about their medical history, lifestyle, and environmental factors. The project has since expanded its data collection to include genotyping and whole-genome sequencing, as well as extensive imaging studies, including brain, heart, and body MRI scans on a subset of 100,000 participants, and physical activity data collected from over 100,000 participants using wrist-worn accelerometers. This comprehensive dataset provides researchers a valuable resource for investigating the causes of a wide range of diseases, with the ultimate aim of improving prevention, diagnosis, and treatment.

### 4.1 SAMPLE SELECTION

We selected our study samples from the UKBB by the following steps: we first identified participants who had both genetic data and accelerometer data, and then we selected individuals diagnosed with PD and paired each PD sample with a healthy control (HC) matched for age, weight, and height. We excluded participants with Alzheimer's Disease or cancer to avoid confounding factors. Since many PD samples lacked sufficient 7-day accelerometer data, we set a minimum requirement of 3 days of data, and applied a quality control step to exclude samples with too many Not-a-Number (NaN) values in their accelerometer data. Finally, participants diagnosed with PD more than two years after the accelerometer data was collected were classified as prodromal PD (PPD) cases. The distribution of the time difference between the year of accelerometer data collection and the year of diagnosis is shown in Figure 2.

## 4.2 PROCESSING

GWAS is a widely-adopted method used to scan the entire genome for genetic variations, particularly SNPs, to identify associations with specific traits or diseases. To identify significant SNPs associated with PD, we conducted GWAS study using PLINK on the imputed genotype dataset from the UKBB (Weeks, 2010). The primary phenotype was PD diagnosis status. To filter out significant SNPs, we applied a p-value threshold of 1e-5, which is commonly used in exploratory GWAS analyses.

LD plays an important role in genetic studies, and therefore we expanded our selection to include the 200 nearest SNPs for each significant SNP, corresponding to approximately 65 kilobases (kb) of genomic distance. This approach allows for a more comprehensive examination of potentially relevant genetic regions. Our analysis yielded 590 significant SNPs from the initial PLINK output. The resulting dataset was structured as a three-dimensional array with dimensions 590 x 200 x 3, where 590 represents the number of significant SNPs, 200 represents the number of nearest SNPs, and 3 represents the probabilities for each possible genotype (homozygous for the reference allele, heterozygous, and homozygous for the alternative allele). We then flattened the last dimension of the array, forming the input $X_{gene} \in \mathbb{R}^{n \times m}$, where $n$ is 590 and $m$ is 600.

Following the Sample Selection and Processing steps, we obtained 215 PD samples, 284 PPD samples, and 499 healthy controls. Each healthy control was paired with either a PD or PPD sample. For data augmentation, we selected a 3-day length for the accelerometer data and applied a 3-hour sliding window to sample the data. The resulting dimension of the accelerometer data, $X_{acc}$, is 51,840.

## 5 RESULTS

Table 1: Model performance comparison for PD vs HC and PPD vs HC classification (mean ± std across 5-fold cross validation)

| Model | PD vs HC | | | PPD vs HC | | |
|---|---|---|---|---|---|---|
| | AUROC (Mean ± Std) | AUPRC (Mean ± Std) | Accuracy (Mean ± Std) | AUROC (Mean ± Std) | AUPRC (Mean ± Std) | Accuracy (Mean ± Std) |
| GRU | 0.907±0.271 | 0.925±0.263 | 0.890±0.367 | 0.805±0.369 | 0.832±0.345 | 0.786±0.368 |
| LSTM | 0.910±0.267 | 0.928±0.250 | 0.898±0.351 | 0.807±0.358 | 0.839±0.338 | 0.792±0.370 |
| Transformer | 0.904±0.280 | 0.923±0.271 | 0.881±0.357 | 0.798±0.373 | 0.820±0.340 | 0.780±0.363 |
| Ours (ACC Only) | 0.918±0.262 | 0.935±0.241 | 0.905±0.352 | 0.814±0.350 | 0.847±0.331 | 0.799±0.361 |
| Ours (ACC + KD) | 0.925±0.246 | 0.940±0.233 | 0.911±0.331 | 0.825±0.341 | 0.854±0.325 | 0.805±0.357 |
| Ours (Fusion) | 0.926±0.241 | 0.943±0.228 | 0.917±0.330 | 0.829±0.342 | 0.859±0.321 | 0.812±0.340 |

## 5.1 IMPLEMENTATION

Our model was implemented using PyTorch 2.0 and trained on a server with 128 GB of memory and an NVIDIA RTX A6000 GPU. The initial training objective was to classify PD from HC, after which the model was fine-tuned to predict PPD from HC. During the training stage, we employed the Adam optimizer with a weight decay of 1e-5. We utilized cosine annealing warm restarts, setting the initial learning rate to 1e-3 and the minimum learning rate to 1e-5 (Loshchilov & Hutter, 2017). For the fine-tuning stage, we froze all model parameters except for the final linear layer, which was kept trainable for fine-tuning. The total number of training epochs was set to 50 and each experiment underwent a 5-fold cross validation.

## 5.2 EVALUATION

As little literature has tried deep learning models for early PD prediction using accelerometer data, to fairly evaluate our model's performance, we conducted comparisons against three prevalent architectures used in time-series data: Gated recurrent unit (GRU), LSTM and Transformer (Cho, 2014). The GRU, LSTM and Transformer models were implemented by replacing Mamba in the ResMamba block with corresponding blocks. This selection of comparative models allows us to assess the efficacy of our approach against both RNN-based and attention-based models.

We selected Area Under the Receiver Operating Characteristic curve (AUROC), AUPRC, and accuracy as our evaluation metrics, as presented in Table 1. For the task of classifying PD from HC, the

Table 2: Genes Related to Parkinson's Disease Discovered by GWAS and GradCAM++ Methods

| GWAS Method | | GradCAM++ Method | |
|---|---|---|---|
| Previously discovered genes | Previously undiscovered genes | Previously discovered genes | Previously undiscovered genes |
| SEPTIN11 | SOS1 | ABCB9 | UVRAG |
| COP1 | AKAP6 | CASC2 | ADAMTS17 |
| SNCA | TTLL13 | ANO10 | LINC00845 |
| GRIK3 | UVRAG | MAPT | GRID2 |
| ANO10 | NEO1 | LINC02210 | GDAP2 |
| DPP6 | SAMD8 | LINGO1 | GPC5 |
| TANC1 | GRAMD2A | FXR1 | AKAP6 |
| LINC02210 | MARCHF4 | SNCA | NECTIN1 |
| KANSL1 | GRID2 | SOX2 | RNF169 |
| PLEK | | ICE1 | SAMD8 |
| FXR1 | | UTRN | ZKSCAN7 |
| MAPT | | DPP6 | TMEM212 |
| SLC17A6 | | | |

Fusion model achieved the best AUROC of 0.926 and AUPRC of 0.943, while in predicting PPD, it achieved an AUROC of 0.829 and AUPRC of 0.859. Notably, all tested models surpassed the previous machine learning model used by Schalkamp et al. (2023), which achieved an AUPRC of 0.78 in both tasks. Among the tested models, the Transformer performed the worst overall, which may be attributed to its known limitation in effectively learning from relatively small datasets. Our ACC-only model outperformed the GRU, LSTM, and Transformer models. Furthermore, the model with KD showed significant improvement compared to the ACC-only model and was comparable to the Fusion model. This indicates that, through KD, the learned knowledge of genetic variants was transferred to the Accelerometer Encoder, thus improving its performance when using ACC data only.

## 5.3 INTERPRETATION

In our interpretation study, we first identified 590 significant SNPs using a p-value threshold of 1e-5 from the GWAS results. We then applied GradCAM++ to investigate which genes our model focuses on (Chattopadhay et al., 2018). Table 2 lists the top 50 genes identified by GWAS and GradCAM++, sorted in descending order of importance based on their respective values. We identified genes that have previously been linked to PD, according to the GeneCards database (Safran et al., 2010). Notably, several genes, including DPP6, SNCA, and MAPT, were identified by both methods and have been previously linked to PD in existing literature (Li et al., 2024; Konno et al., 2016; Zabetian et al., 2007). Additionally, we found genes such as AKAP6, UVRAG, SAMD8, and GRID2 that were highlighted by both GradCAM++ and GWAS but have been less frequently associated with PD in previous studies. Among these findings, SOS1 and UVRAG emerged as the top genes identified by GradCAM++ and GWAS, respectively. UVRAG, a key regulator of autophagy and endosomal trafficking, may contribute to PD pathogenesis through impaired clearance of protein aggregates and dysfunctional mitochondria, potentially exacerbating neuronal dysfunction and death (Yin et al., 2011). The SOS1 gene, encoding a guanine nucleotide exchange factor for RAS proteins, may contribute to PD through its involvement in EGFR-mediated neuroprotective signaling pathways and potential interaction with LRRK2, a major genetic risk factor for the disease (Chardin et al., 1993). The combined effects of UVRAG's role in cellular quality control and SOS1's influence on neuroprotective signaling pathways may represent a novel axis in the complex molecular landscape of PD, offering potential targets for therapeutic intervention and biomarker development.

# 6 CONCLUSIONS

In this paper, we introduced GeneMamba, a model integrating genetic and accelerometer data for the early prediction of PD. Our model achieved an AUPRC of 0.943 in classifying PD subjects from HC, and an AUPRC of 0.859 in predicting PPD cases up to 7 years before clinical diagnosis. By employing a knowledge transfer approach, we enhanced the performance of our model utilizing accelerometer data only, which is easier to obtain through wearable devices and has greater real-world applicability. Furthermore, our GWAS analysis revealed several significant genes associated with PD, while GradCAM++ provided insights into the genes prioritized by our model. These results were consistent with previous studies, confirming the importance of several genes already known to be significant in PD. Additionally, our methods identified genes that have been less frequently associated with PD in previous research, namely UVRAG and SOS1, which emerged as top genes from GradCAM++ and GWAS analyses, respectively. These findings suggest potential new avenues for PD prevention, although further investigation is required to confirm their relevance to the disease. Overall, GeneMamba represents a novel method for the early detection of PD, combining accelerometer and genetic data to improve PD prediction and potentially uncover novel genetic factors associated with PD. This approach could contribute significantly to early intervention strategies and personalized medicine in the field of neurodegenerative diseases.

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
