# OpenReview forum: "GeneMamba: Early Parkinson’s Detection via Wearable Device and Genetic Data"
_ICLR.cc/2025/Conference — Submitted to ICLR 2025_

### Official Review · Reviewer_kCDM · 2024-10-27

**Soundness:** 2
**Presentation:** 2
**Contribution:** 1
**Rating:** 3
**Confidence:** 4

**Summary:**

The paper proposes a multimodal Structured State Space Model, GeneMamba, for early detection of Parkinson's disease (PD). Two modality is studied here (1) accelerometer data and (2) gene data.
GeneMamba is a late-fusion model using separate Accelerometer Encoder and Gene Encoder to extract features. The fusion module is a cross-attention layer.
The Accelerometer Encoder uses a combination of Mamba and 1D Convolution blocks.
The Gene Encoder uses a combination of LSTM and 2D Convolution blocks.
The model is evaluated on two classification tasks PD vs healthy and Prodromal PD vs healthy.

**Strengths:**

Using Mamba to analyze accelerometer data is novel.

**Weaknesses:**

1) Questionable baselines. The authors select an arbitrary architecture backbone and then replace the Mamba blocks with GRU, LSTM, Transformer blocks. How do the authors ensure that the selected architecture is the best for each new block and not overly optimized for Mamba. For example, when using the Transformer blocks, the authors could decrease the number of layers to prevent overfitting on the small dataset?
2) Lack of comparison with current works and multimodal baselines. I do not agree with the author's claims that "little literature has tried deep learning models for early PD prediction using accelerometer data". Please see [1, 2, 3] for some references. In addition, there are early, late, and joint fusion approaches which are well-understood and common baselines for dealing with multimodal healthcare data [4].
3) The discussion on the novel genes discovered is unsubstantiated. Although the model selects some unidentified genes, there is no guarantee that it is relevant to PD pathology.
4) Lack of implementation details in terms of model architecture (e.g., number of layers, hidden layer size etc) and loss function. There are some descriptions in the main text, but uses vague language (e.g., We gradually increase the size of the feature embeddings from 64 to 512). The loss function has two separate loss components and an α term is used in equation 3 to weigh the losses. However, the authors do not provide the α value that is used.

[1] Prince, J., Andreotti, F., & De Vos, M. (2018). Multi-source ensemble learning for the remote prediction of Parkinson's disease in the presence of source-wise missing data. IEEE Transactions on Biomedical Engineering, 66(5), 1402-1411.

[2] Schwab, P., & Karlen, W. (2019, July). PhoneMD: Learning to diagnose Parkinson’s disease from smartphone data. In Proceedings of the AAAI conference on artificial intelligence (Vol. 33, No. 01, pp. 1118-1125).

[3] Li, W., Zhu, W., Dorsey, E. R., & Luo, J. (2020, November). Predicting Parkinson's Disease with Multimodal Irregularly Collected Longitudinal Smartphone Data. In 2020 IEEE International Conference on Data Mining (ICDM) (pp. 1106-1111). IEEE.

[4] Huang, S. C., Pareek, A., Seyyedi, S., Banerjee, I., & Lungren, M. P. (2020). Fusion of medical imaging and electronic health records using deep learning: a systematic review and implementation guidelines. NPJ digital medicine, 3(1), 136.

**Questions:**

1) How is the model backbone used in the experiments selected?
2) How does the proposed approach compare with approaches stated in the Weaknesses section?
3) Why is LSTM used in Gene Encoder instead of Mamba block?
4) How important is the gene modality? What is the accuracy on the PD vs HC and PPD vs HC when using the gene encoder only?
5) How does the gene discovered change when using GeneMamba vs GeneEncoder only. Does using GeneEncoder only also uncover unidentified genes?

---

### Official Review · Reviewer_Qtyy · 2024-10-31

**Soundness:** 2
**Presentation:** 2
**Contribution:** 2
**Rating:** 5
**Confidence:** 4

**Summary:**

The paper presents GeneMamba, a multi-modal deep learning model aimed at early detection of Parkinson's Disease (PD) by integrating accelerometer data from wearable devices and genetic data. Using a structured state space model, GeneMamba predicts PD onset up to seven years in advance. The authors employ cross-modality fusion for data integration and knowledge distillation to transfer insights from genetic data to accelerometer-only models. Additionally, the model uses interpretability methods(GWAS and GradCAM++) to identify PD-associated genes, offering potential new targets for therapeutic intervention.

**Strengths:**

1) Multi-modal Integration for Enhanced Prediction: GeneMamba’s integration of genetic and accelerometer data provides a robust approach for early PD prediction, with significant predictive accuracy (AUPRC 0.943 for PD vs. HC).
2) Interpretability and Novel Gene Identification: Using GradCAM++ and GWAS, the model highlights genes (e.g., UVRAG and SOS1) previously unassociated with PD.
3) State Space Modeling for Time-Series Data: The choice of Mamba, a Structured State Space Model, allows efficient handling of long time-series data. (Proposed by Albert gu , rejected in ICLR)

**Weaknesses:**

1) Lack of Table Related to Model's Layer Details: The absence of a hyperparameter table, including layer sizes and configurations (such as ResMamba block specifications), affects reproducibility
2) Freezing Layer Comparison: Te model’s fine-tuning involved freezing all layers except the final one. However, the effects of freezing additional layers (e.g., last two or three) on performance and stability remain unexplored, limiting insights into how deeper layer freezing might affect knowledge transfer and model robustness.
3) Limited Evaluation Across Datasets: The model’s evaluation is limited to the UK Biobank dataset, so less generalizability , might use  AMP MDS-UPDRS dataset.

**Questions:**

1) How does freezing multiple layers impact performance?
2) Would the model perform similarly on other datasets related to Parkinson's, such as the amp mds updrs dataset?
3) Could the authors provide a detailed table about how many layers are there and their shape?
 4) Could the authors elaborate on how the cross-attention fusion is implemented within the Mamba SSM, especially given that Mamba’s framework avoids traditional attention?

---

### Official Review · Reviewer_X94j · 2024-11-02

**Soundness:** 2
**Presentation:** 3
**Contribution:** 2
**Rating:** 3
**Confidence:** 2

**Summary:**

This paper proposes a model called GeneMamba that uses genetic data and accelerometer signals to do better performance in predictive tasks. What is novel about GeneMamba is the use of a state-space model that integrates cross-attention mechanisms to fuse together genetic and sensor-based information. The approach is introduced here as an alternative to traditional non-attention-based mechanisms. The authors present significant improvements both in terms of selectivity and generalizability with experimental evidence on a single dataset. However, their level in hyperparameters, layer configurations, and the data augmentation methods have not been wholly covered.

**Strengths:**

Originality: GeneMamba brings out the cross-attention-based SSM that in itself will introduce a creative fusion of genetic and sensor data that may open new avenues for multi-modal data integration.

Clarity: The paper is largely well structured and explains most concepts in clear language.

Potential Impact: Such an integration of genetic and accelerometer data would be likely to have a relevant
impact upon this type of application and the demands of personalized medicine and health diagnostics, which rely heavily on multifaceted approaches.

**Weaknesses:**

Reproducibility Issues: There is a lack of key implementation details including hyperparameters, configurations of the layers (e.g. ResMamba block) and data augmentation. Such details would enhance reproducibility and be much useful to the research community in general.

Limited Evaluation: Testing on only one dataset limits the paper's insights into the model's generalizability. Additional datasets, such as MODMA or MDS-UPDRS, could offer more robust evidence for GeneMamba's applicability across diverse data distributions.

Cross-Attention Mechanism: GeneMamba's description of their cross-attention mechanism is rather shallow. It is not clear at all how that implementation would compare to an attention model of normal types and how the selectivity mechanism of the model is impacted by that approach.

Layer Freezing Experimentation: Freeze multiple layers without detailed study. Experiment different layer-freezing configurations such as freezing the last two or three layers to elucidate the stability-flexibility fine-tuning trade-offs.

**Questions:**

Could you clarify how the Mamba SSM replaces the conventional non-attention-based mechanism in Aiven Mamba? Specifically, how is cross-attention implemented here, and does it influence Mamba’s selectivity mechanism?

Would you be able to provide a comprehensive hyperparameter table? Including details on layer sizes, configurations (e.g., ResMamba block), and data augmentation techniques would significantly enhance the work’s reproducibility.

How does freezing multiple layers affect performance? Comparisons of freezing only the last layer versus additional layers would provide insights into balancing flexibility with stability.

Have you considered evaluating the model on other datasets, such as MODMA or MDS-UPDRS, to assess its generalizability? This would help address any dataset-specific biases.

---

### Meta-Review · Area_Chair_18ve · 2024-12-21

**Metareview:**

This paper was reviewed by three experts in the field and received 3, 5, 3 as the ratings; it presents GeneMamba, a multi-modal deep learning model for early detection of Parkinson’s Disease (PD), by integrating accelerometer data from wearable devices and genetic data. The reviewers agreed that such a technique can have potential impacts on personalized medicine and health diagnostics, the paper is well-structured and the ideas are clearly expressed, for the most part, and the experimental results are encouraging.

The reviewers raised concerns about the lack of comparison with current state-of the-art methods. It is difficult to assess the merit of GeneMamba without an exhaustive comparison against the relevant techniques. Further, the proposed method has been validated only on a single dataset; it needs to be validated on multiple datasets to appropriately understand its usefulness and generalizability across different data distributions. It was also mentioned that although the model selects some unidentified genes, there is no guarantee that they are relevant to PD pathology; further discussion and validation are necessary. Concerns were also raised about the reproducibility of the proposed method, as several key implementation details have not been provided, such as hyperparameters, configurations of the layers (e.g. ResMamba block) and data augmentation.

The authors did not respond to the individual reviewer's comments. In light of the above discussions, we conclude that the paper is not ready for an ICLR  publication in its current form. While the paper clearly has merit, the decision is not to recommend acceptance. The authors are encouraged to consider the reviewers' comments when revising the paper for submission elsewhere.

**Additional Comments On Reviewer Discussion:**

Please see my comments above.

---

### Decision · Program_Chairs · 2025-01-22

Reject